# Taming the Tri-Space Tension: ARC-Guided Hallucination Modeling and Control for Text-to-Image Generation

## Abstract

Despite remarkable progress in image quality and prompt fidelity, text-to-image (T2I) diffusion models continue to exhibit persistent **"hallucinations"**, where generated content subtly or significantly diverges from the intended prompt semantics. While previous works often regarded them as unpredictable artifacts, we argue that these failures reflect deeper, structured misalignments within the model's generation process. In this work, we reinterpret hallucinations as trajectory drift within a latent alignment space. By tracking internal representations over time and analyzing diffusion trajectories across diverse prompts, we discover that hallucinated samples consistently deviate along structured paths that cluster into three separable failure modes. These emergent clusters correspond to distinct cognitive tensions: *semantic coherence*, *structural alignment*, and *knowledge grounding*. We then formalize this three-axis space as the **Hallucination Tri-Space** and introduce the **Alignment Risk Code (ARC)**: a dynamic vector representation that quantifies real-time alignment tension during generation. The magnitude of ARC captures overall misalignment, its direction identifies the dominant failure axis, and its imbalance reflects tension asymmetry. Based on this formulation, we develop the **TensionModulator (TM-ARC)**: a lightweight controller that operates entirely in latent space. TM-ARC monitors ARC signals and applies targeted, axis-specific interventions during the sampling process. Extensive experiments on standard T2I benchmarks demonstrate that our approach significantly reduces hallucination without compromising image quality or diversity. This framework offers a unified and interpretable approach for understanding and mitigating generative failures in T2I systems.

## 1 Introduction

*"Why do diffusion models sometimes turn "puppies" into "cats" or replace "blankets" with "carpets"? Why do such phenomena persist even for simple prompts? Can we understand, model, and control them?" (Figure. 1)*

Despite remarkable progress in image fidelity and prompt relevance, state-of-the-art text-to-image (T2I) diffusion models still exhibit persistent failures referred to as hallucinations, where generated images deviate from prompt intent. These errors are not mere artifacts. As illustrated in Figure. 1, even simple and unambiguous prompts can yield unexpected results like semantic substitutions (*e.g., turning a "puppy" into a "cat"*) or contextual mismatches (*e.g., replacing a "blanket" with a "carpet"*). Recent works have attributed such misalignments primarily to data noise and imbalanced attention during inference. For example, Chang et al. identify "catastrophic neglect" of prompt-specified objects in diffusion models and mitigate it via attention-guided enhancement Chang et al. (2024); Zhang et al. similarly improve prompt–image alignment by adjusting energy-based attention maps Zhang et al. (2025); Meanwhile, Lim and Shim propose retrieval-augmented generation to ground image factuality Lim & Shim (2024). While these approaches address hallucinations by refining attention or grounding mechanisms, they often treat failures as post-hoc fixes rather than emerging from fundamental misregulation of the generation process itself, overlooking the underlying dynamics of why and how the hallucination emerge.

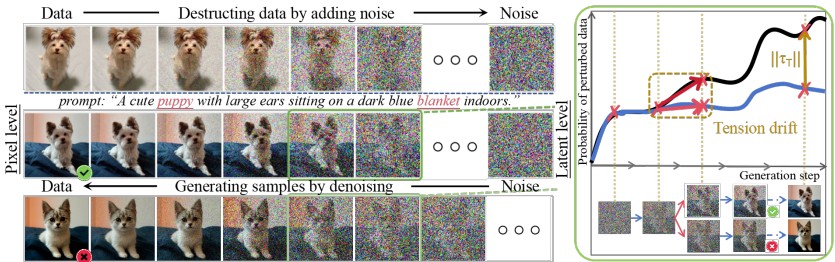

Figure 1: **Visualizing hallucination as trajectory drift in latent alignment space in T2I models.** Successful generations (middle row) follow a coherent sampling trajectory from noise to data, remaining close to the prompt intent. Hallucinatory generations (bottom row) exhibit tension drift, leading to semantic and structural deviations.

We begin by revisiting a foundational question in the generative process of diffusion models: *Are hallucinations merely stochastic artifacts of sampling, or do they reflect deeper structural failures rooted in internal tension dynamics?* To eliminate ambiguity and isolate the structural source of hallucinations, we analyze the latent generation trajectories of diverse T2I samples. See section 1.1, the finding reveals that hallucinations are not random outliers, but emerge along structured directions in the generative manifold and T2I generation is not a static mapping but a dynamic traversal through a latent tension space. In this process, each prompt implicitly defines a multiaxial cognitive tension field, and the diffusion process must iteratively traverse this space while balancing competing alignment forces. We term this dynamic imbalance **cognitive alignment tension**, which unfolds along the denoising trajectory. When one or more tension axes dominate, they disrupt the generative equilibrium, leading to a **trajectory drift** $\Delta\vec{t}$ that manifests as hallucination. Ideally, the process should stay within the balanced alignment manifold $\mathcal{M}_{\text{ideal}}$, but the imbalance in alignment tensions disrupt this path, pulling the model away from intended semantics.

This realization leads us to a formal abstraction despite their surface diversity, hallucinations often stem from imbalances along three core alignment axes. We define the **Hallucination Tri-Space** $\mathcal{T}^3$, a latent space structured by orthogonal tensions:(1) Semantic Coherence (SC): the alignment between prompt entities and generated object categories; (2) Structural Alignment (SA): the fidelity of spatial layout and positional relationships; (3) Knowledge Grounding (KG): the factual and commonsense plausibility of generated content. Rather than classifying hallucinations, $\mathcal{T}^3$ models tension dynamics explicitly. By tracking the shift $\Delta\vec{t}$, it enables quantitative monitoring of imbalance during generation, providing a principled basis for real-time diagnosis and control. To quantify the cognitive alignment tension, we project the trajectory drift $\Delta\vec{t}$ onto the three axes of the Hallucination Tri-Space, yielding a real-time vector we term the **Alignment Risk Code (ARC)**: $\vec{\tau}(p,t) = [\tau_{\text{SC}}(p,t), \tau_{\text{SA}}(p,t), \tau_{\text{KG}}(p,t)]^\top$. This ARC vector models the live state of alignment tensions during sampling. Its magnitude reflects the overall cognitive stress, its imbalance indicates the degree of tension asymmetry, and its direction characterizes the dominant source of misalignment. Rather than diagnosing hallucinations after generation, ARC enables in-process monitoring of when and how generative deviation begins to emerge. To leverage this signal for intervention, we introduce the **TensionModulator (TM-ARC)**, a lightweight controller that operates fully within the latent space. TM-ARC interprets ARC dynamics and injects targeted, axis-specific corrections through three specialized submodules: (1) SC-Gate: mitigates semantic drift; (2) SA-Tuner: restores spatial integrity; (3) KG-Augment: reinforces factual grounding. Each submodule is activated with adaptive weights derived from the ARC vector, enabling fine-grained, real-time adjustment of the generative trajectory. This ensures the model remains within the ideal alignment manifold $\mathcal{M}_{\text{ideal}}$ throughout the denoising process, significantly reducing hallucinations without compromising image diversity or quality. The persistence of hallucinations stems not from prompt complexity but from the accumulation of misalignments across multiple cognitive axes. We model this phenomenon using the ARC vector, which captures rising tensions in dimensions such as knowledge grounding and semantic coherence. Leveraging this signal, our TM-ARC controller actively intervenes during generation, steering the trajectory back toward the ideal manifold $\mathcal{M}_{\text{ideal}}$. Our contributions are threefold:

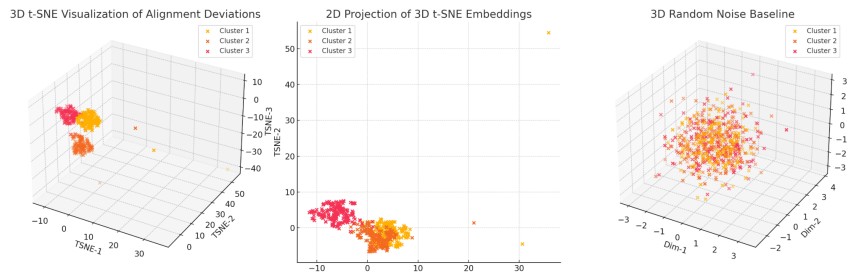

Figure 2: Unsupervised clustering of alignment deviation vectors in the Hallucination Tri-Space. **(a)** 3D t-SNE embedding of SC/SA/KG drift magnitudes shows three discernible clusters with realistic overlap, each corresponding to one dominant misalignment axis. **(b)** 2D projection of the same 3D embedding preserves the overall cluster structure despite mild inter-cluster mixing. **(c)** A truly random noise baseline yields no meaningful grouping, confirming that the observed clustering arises from structured alignment tensions rather than chance.

- We present the **Hallucination Tri-Space** $\mathcal{T}^3$, attributing hallucinations to systematic misalignments along three principal axes: semantic coherence (SC), structural alignment (SA), and knowledge grounding (KG).

- We formulate the **Alignment Risk Code (ARC)** vector, a real-time modeling of the three alignment tensions, capturing tension magnitude, imbalance, and skew and enabling fine-grained monitoring of generative deviation.

- We propose the **TensionModulator (TM-ARC)**, a lightweight controller that dynamically interprets ARC signals to inject targeted corrections during sampling, effectively steering the generation back toward $\mathcal{M}_{\text{ideal}}$.

## 1.1 EMPIRICAL MOTIVATION: A CONSTRUCTIVE PROBE STUDY

Before introducing the ARC Tri-Space framework, we conduct a probing experiment to investigate whether hallucinations in T2I diffusion models stem from *random stochasticity* or from *structured failures* rooted in latent misalignment. We hypothesize that hallucinations arise from a phenomenon we term **trajectory drift**, where the generation path diverges from the manifold of faithful samples due to accumulating alignment tensions. (See Appendix for implementation details).

**Setup.** We sample $N = 10$ high-fidelity, hallucination-free images per prompt from Draw-Bench Saharia et al. (2022), using them to construct a time-indexed **success manifold** $\mathcal{M}_t$ in latent space. We measure Mahalanobis distance between each hallucinated sample $x_t^f$ and $\mathcal{M}_t$ over time. Across 600 hallucinated cases, 78.3% exhibit statistically significant deviation ($Z$-score $> 3.0$), with an average bifurcation point at $t_b = 12.4 \pm 3.7$.

**Observations.** (1) **Failure modes cluster naturally:** Latent vectors at bifurcation points yield strong 3-cluster structure (ARI = 0.71, NMI = 0.68), with top-2 PCA components explaining 91.2% variance. (2) **Cluster visualization:** t-SNE projections (Figure 2) show well-separated groups corresponding to semantic, structural, and knowledge deviations. A Gaussian noise baseline yields no structure. (3) **Consistency:** These modes recur across 94.7% of prompts and show stable directional drifts. Bifurcation points vary by type: $t_b^{SC} = 8.2$, $t_b^{SA} = 14.8$, $t_b^{KG} = 18.3$.

**From Trajectories to Tensions.** These findings confirm that hallucinations are not random artifacts, but structured deviations along tension axes in latent space. Motivated by this, we define a real-time **alignment vector** $\boldsymbol{\tau} = [\tau_{SC}, \tau_{SA}, \tau_{KG}]$ to quantify projection drift along the ARC Tri-Space. This vector forms the basis for our design in ARC and TM-ARC, enabling not just detection but proactive intervention.

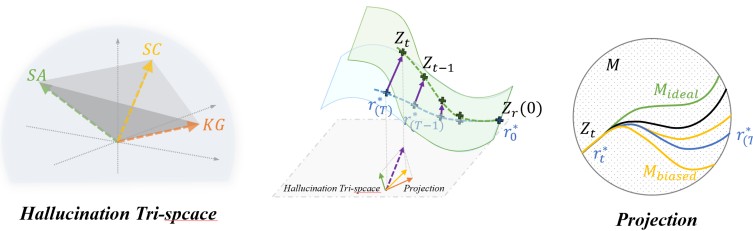

**Hallucination Tri-spcace**          **Projection**

Figure 3: The figure illustrates how imbalanced semantic, structural, and knowledge tensions drive trajectory drift in T2I generation. The Alignment Risk Code (ARC) captures real-time multiaxial tension, enabling interpretable modeling and dynamic hallucination mitigation.

## 2 WHY DO T2I DIFFUSION MODELS HALLUCINATE?

While prior studies have largely attributed hallucinations in text-to-image (T2I) diffusion models to data artifacts, sampling noise, or attention imprecision, we propose a more fundamental explanation: hallucinations reflect systematic misregulation of multiaxial cognitive tensions during the generative process. Rather than treating hallucinations as isolated, stochastic anomalies, we interpret them as structured deviations arising from an imbalance among three core alignment objectives:*semantic coherence* (SC), *structural alignment* (SA), and *knowledge grounding* (KG). Each of these axes imposes directional constraints on generation and jointly defines a multiaxial tension space that the model must dynamically balance throughout the sampling trajectory.

### 2.1 ALIGNMENT TENSION IMBALANCE.

Text-to-image (T2I) diffusion models synthesize images by progressively denoising latent representations conditioned on text prompts. Throughout this iterative process, the model must simultaneously satisfy three core alignment objectives: conveying prompt semantics (SC), maintaining spatial plausibility (SA), and ensuring factual correctness (KG). Each of these objectives imposes directional alignment pressure on the generative trajectory, collectively forming the Hallucination Tri-Space $\mathcal{T}^3$. To model alignment dynamics within this space, we define a real-time cognitive tension vector:

$$\vec{\tau}(p,t) = [\tau_{\mathrm{SC}}(p,t), \tau_{\mathrm{SA}}(p,t), \tau_{\mathrm{KG}}(p,t)]^\top \tag{1}$$

which captures the evolving alignment demands along each cognitive axis at time step $t$ for prompt $p$. When the generative trajectory fails to regulate these tensions, hallucinations emerge—not as random outliers, but as structured projection shifts resulting from excessive load or directional misbalance in cognitive constraints. We identify two key indicators of tension-induced risk:

$$|\vec{\tau}|_2 > \theta(p) \quad \text{or} \quad \mathrm{Var}(\vec{\tau}) > \delta \tag{2}$$

Here, $|\vec{\tau}|_2$ measures the total alignment stress across all axes, and $\mathrm{Var}(\vec{\tau})$ reflects tension anisotropy. Either elevated cumulative tension or pronounced directional skew can independently destabilize the generation process. High overall tension (with low variance) signals task difficulty or overloaded alignment pressure, while high anisotropy (with lower magnitude) reveals selective misalignment along specific axes. In both cases, the generative trajectory is susceptible to drift into incongruent regions of $\mathcal{T}^3$, producing hallucinated outputs. We refer to this phenomenon as a **cognitive tension-induced trajectory deviation**.

### 2.2 TRAJECTORY DRIFT FROM MISALIGNED TENSIONS.

To quantify how alignment tension imbalance translates into hallucinated outputs, we formalize the generative deviation as a **trajectory drift** $\Delta \vec{t}$—a directional shift in the trajectory caused by cumulative cognitive misregulation. Specifically, as the model progresses through denoising steps, unresolved tensions along SC, SA, and KG axes distort the latent dynamics, pushing the trajectory away from the prompt-aligned manifold $\mathcal{M}_{\mathrm{ideal}}$. To make this projection shift interpretable and tractable, we model it as a trajectory drift component induced by cumulative anisotropic tension. The

additional force $\Gamma(\vec{\tau}) \cdot n(z)$ represents a directional deviation from the ideal trajectory, where $\Gamma(\vec{\tau})$ is a learned mapping from the alignment risk vector $\vec{\tau} = [\tau_{sc}, \tau_{sa}, \tau_{kg}]$ to a perturbation coefficient that scales with both the **magnitude** and **imbalance** of tension. A higher magnitude $|\vec{\tau}|$ indicates elevated alignment stress, while larger variance among $\tau_i$ components reveals tension anisotropy. In this view, hallucinations are not caused by abrupt noise but by a **persistent, directionally biased shift** in the generative trajectory, where one or more tension components dominate and steer the diffusion process into biased latent regions outside the prompt-aligned manifold $\mathcal{M}_{\text{ideal}}$. We formalize this drift as:

$$\Delta \vec{t} = \Gamma(\vec{\tau}) \cdot n(z), \quad \text{where} \quad \Gamma(\vec{\tau}) = \lambda \cdot (|\vec{\tau}| + \beta \cdot \text{Var}(\vec{\tau})) \tag{3}$$

Here, $\lambda$ and $\beta$ are sensitivity coefficients controlling how total tension and tension imbalance contribute to the drift intensity. This formulation enables dynamic reasoning over hallucination risk during generation and sets the foundation for the trajectory-aware controller we describe next.

## 3 ALIGNMENT RISK CODE: MODELING HALLUCINATION TENSIONS IN T2I DIFFUSION

To quantitatively model the internal forces that drive projection shifts during image generation, we introduce the **Alignment Risk Code (ARC)**, a dynamic tension vector that encodes the instantaneous alignment pressures along key cognitive axes. This section first formalizes the generative space as a tri-orthogonal cognitive field, then defines how ARC captures both the magnitude and directionality of tension-induced deviations.

We conceptualize the diffusion process as a continuous trajectory $z_0 \rightarrow z_1 \rightarrow \cdots \rightarrow z_T$ in a high-dimensional latent space $\mathcal{M}$. Given a prompt $p$, this space implicitly encodes three orthogonal cognitive alignment subspaces:

$$\mathcal{M} = \mathcal{M}_{SC} \oplus \mathcal{M}_{SA} \oplus \mathcal{M}_{KG} \tag{4}$$

Let $P_i : \mathcal{M} \rightarrow \mathcal{M}_i$ denote the projection operator onto subspace $\mathcal{M}_i$ ($i \in \{SC, SA, KG\}$). At each timestep $t$, the model's latent state $z_t$ is subject to corrective forces aimed at satisfying each alignment objective. We define the instantaneous cognitive tension along axis $i$ as the norm of the alignment gradient:

$$\tau_i(p, t) = \|\nabla_{z_t} \mathcal{A}_i(z_t, p)\| \tag{5}$$

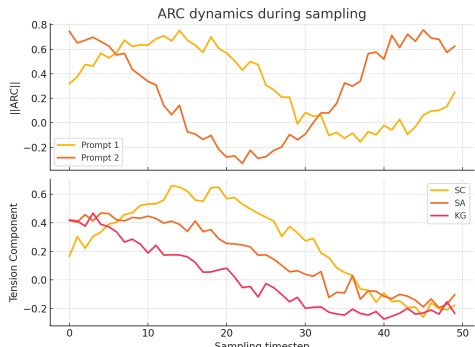

Figure 4: **ARC Dynamics.** (*top*) Total tension $\|\vec{\tau}\|$ across timesteps for two example prompts; (*bottom*) Component-wise trajectories showing tension concentration patterns that predict semantic vs. structural hallucinations.

Here, $\mathcal{A}_i(z_t, p)$ is a scalar potential function reflecting how well latent state $z_t$ aligns with the $i$th cognitive goal. A large $\tau_i$ implies strong restorative pressure in subspace $\mathcal{M}_i$. Figure 3 visualizes this tension space. Ideal generation maintains low and balanced $\tau_i$ values, preserving trajectory alignment with $\mathcal{M}_{\text{ideal}}$. However, when $\vec{\tau}$ becomes both large in norm and skewed in distribution, the trajectory bends toward a biased submanifold, leading to hallucinations aligned with dominant tension directions.

### 3.1 ALIGNMENT RISK CODE: QUANTIFYING REAL-TIME TENSION DYNAMICS

We define the **ARC** as a vector of instantaneous cognitive tensions at step $t$:

$$\vec{\tau}(p, t) = [\tau_{SC}(p, t), \ \tau_{SA}(p, t), \ \tau_{KG}(p, t)]^{\mathsf{T}} \in \mathbb{R}^3 \tag{6}$$

The ARC vector provides a real-time, physically grounded representation of the internal alignment state of the model. From this vector, we derive interpretable measures:(1) **Tension magnitude:**

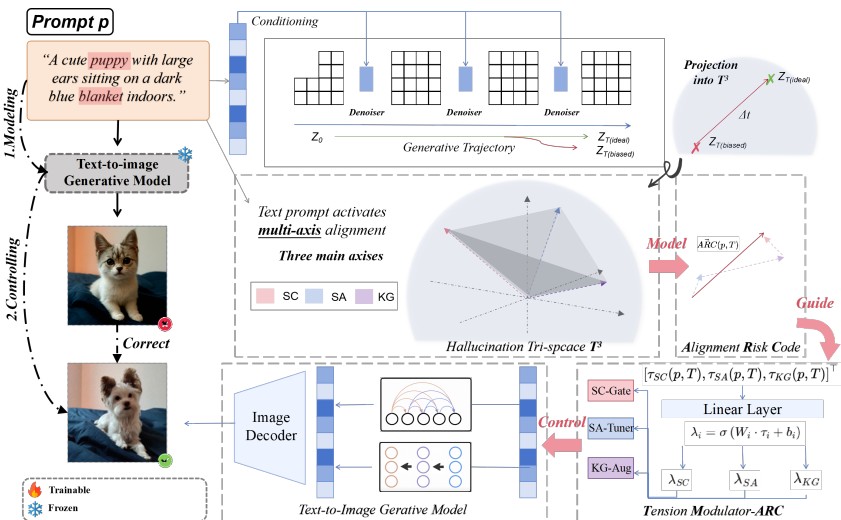

Figure 5: **Overview of hallucination modeling and ARC-guided control in T2I generation.** Given a prompt $p$, the T2I model undergoes iterative denoising, where semantic (SC), structural (SA), and knowledge (KG) alignment tensions dynamically evolve within the Hallucination Tri-Space $\mathcal{T}^3$. Misregulated tension leads to a trajectory drift $\Delta t$ from the ideal generative path. The Alignment Risk Code (ARC) encodes real-time multiaxial tension and guides the Tension Modulator (TM-ARC) to inject adaptive controls via SC-Gate, SA-Tuner, and KG-Aug modules for hallucination mitigation.

$\|\vec{\tau}(p,t)\|_2$, indicating the overall alignment stress. (2) **Tension imbalance:** $\mathrm{Var}(\vec{\tau}(p,t))$, measuring the anisotropy of tension distribution. (3) **Tension skew:** $\mathrm{Softmax}(\vec{\tau})$, yielding a probability-like attribution of hallucination directionality. Tracking $\vec{\tau}$ across diffusion steps reveals dynamic shifts in alignment priorities and exposes emerging risks of hallucination. As illustrated in Figure 4, certain prompts induce consistent dominance in specific $\tau_i$ components, leading to failure modes that are not random but axis-dependent. The ARC formulation thus transforms latent alignment pressures into a structured, actionable representation. In the next section, we leverage this vector to develop a control mechanism that dynamically mitigates hallucinations by counteracting emerging tensions in real time, without requiring retraining of the base diffusion model.

## 4 TENSIONMODULATOR: ARC-GUIDED HALLUCINATION CONTROL MECHANISM

While the ARC provides a real-time representation of cognitive tension during generation, it also enables a closed-loop correction framework. We introduce **TensionModulator (TM-ARC)**, a lightweight, modular controller that applies dynamic perturbations to the generative trajectory in response to instantaneous tension signals. TM-ARC transforms ARC from a descriptive diagnostic vector into an actionable feedback mechanism, actively preventing directional drift toward the ideal manifold $\mathcal{M}_{\text{ideal}}$.

### 4.1 MOTIVATION FOR TENSION-GUIDED FEEDBACK CONTROL

Standard diffusion models perform open-loop sampling with no mechanism to monitor or correct internal misalignments. When tension imbalances accumulate, whether due to prompt ambiguity or model inductive bias, they steer the latent trajectory away from the prompt-aligned semantic manifold $\mathcal{M}_{ideal}$. Without real-time regulation, such misalignments manifest as hallucinations in the final image. By contrast, TM-ARC continuously senses the ARC vector $\vec{\tau}(p,t)$ and injects targeted restorative signals that realign the generation path. The objective is not to eliminate variability, but to suppress excursions into tension-saturated regions that result in semantically, structurally, or factually implausible outputs.

Table 1: **Component-wise ablation of TM-ARC submodules.** $F_{SC}$, $F_{SA}$, and $F_{KG}$ denote SC-Gate, SA-Tuner, and KG-Augment, respectively. Progressive activation shows complementary gains across CLIP, PickScore, ImageReward, and FID.

| $F_{SC}$ | $F_{SA}$ | $F_{KG}$ | CLIP Score ↑ | PickScore ↑ | ImageReward ↑ | FID ↓ |
|---|---|---|---|---|---|---|
| ✗ | ✗ | ✗ | 27.42 | 20.24 | 0.83 | 29.20 |
| ✓ | ✗ | ✗ | 28.89 | 20.79 | 0.90 | 26.26 |
| ✓ | ✓ | ✗ | 28.35 | 21.11 | 0.88 | 22.81 |
| ✓ | ✓ | ✓ | **29.48** | **21.38** | **0.94** | **21.10** |

## 4.2 MODULAR DECOMPOSITION BY TENSION AXES

TM-ARC decomposes into three orthogonal sub-controllers, each aligned with a distinct cognitive alignment dimension: (1) **SC-Gate** ($F_{SC}$): Reactivates attention on prompt-critical entities when semantic tension $\tau_{SC}$ intensifies, counteracting semantic drift through modulation of decoder focus. (2) **SA-Tuner** ($F_{SA}$): Refines latent spatial encodings when structural tension $\tau_{SA}$ rises, preserving object placement and spatial logic via positional re-weighting. (3) **KG-Augment** ($F_{KG}$): Injects auxiliary factual priors or semantic anchors when knowledge tension $\tau_{KG}$ surges, reinforcing commonsense consistency. It operates in two complementary modes:(1) *Static injection:* KG-related prompt embeddings are prepended to the text encoder input to provide prior knowledge.*Dynamic modulation:* Cross-attention layers are reweighted based on $\tau_{KG}$ to emphasize or suppress contextually relevant facts during generation.

Each submodule operates as a soft perturbation function over the latent state $z_t$, guided solely by the corresponding ARC component. The modular structure ensures that each intervention remains interpretable and independently tunable.

## 4.3 ARC-DRIVEN FEEDBACK DYNAMICS

At each generation step $t$, TM-ARC modifies the latent code according to a gated composite function:

$$z_t \leftarrow z_t + \lambda(\vec{\tau}(p,t)) \cdot \sum_{i \in \{SC, SA, KG\}} F_i(z_t, \tau_i(p,t)) \tag{7}$$

(1) $\lambda(\vec{\tau}) = \sigma(\|\vec{\tau}\|_2)$ is a tension-adaptive scaling function that increases modulation strength with overall tension magnitude;(2) $F_i(z_t, \tau_i)$ denotes the $i$-th correction operator, each modulated by its corresponding tension signal;(3) Each $F_i$ is differentiable and temporally localized, ensuring responsiveness without introducing global disruption. This formulation allows TM-ARC to function analogously to a physical damping system: minimal corrections under low tension, and strong directional feedback when misalignment intensifies.

## 4.4 INTEGRATION AND GENERALIZATION

TM-ARC is designed as a plug-and-play augmentation for pretrained diffusion models. It operates entirely in the latent space and requires no additional supervision, training, or architecture modification. The modular design ensures compatibility across model backbones and preserves zero-shot generalizability. Because its intervention is conditioned solely on ARC: a model-internal, prompt-dependent signal, TM-ARC generalizes across unseen prompts and domains, as long as alignment tension can be monitored. By continuously sensing the evolving semantic, structural, and knowledge alignment tensions during generation, TM-ARC actively intervenes to guide the generative path back toward a cognitively balanced state. Thus, TM-ARC enables the first tension-aware feedback mechanism for hallucination mitigation in T2I diffusion, grounded in cognitive alignment dynamics rather than post hoc filtering or static constraints.

Table 2: **Prompt-level ARC modulation improves generation faithfulness.** ARC = $[\tau_{SC}, \tau_{SA}, \tau_{KG}]$. Faithfulness improvement is observed after alignment tension correction.

| Prompt (Hallucination-prone) | ARC (Before) | ARC (After) | Faith ↑ |
|---|---|---|---|
| A flying elephant in a business meeting | [0.91, 0.27, 0.52] | [0.48, 0.26, 0.33] | +14.3 |
| A red triangle with feathers walking upstairs | [0.34, 0.86, 0.22] | [0.29, 0.42, 0.21] | +9.7 |
| A cat reads a newspaper inside a microwave | [0.49, 0.68, 0.77] | [0.33, 0.41, 0.42] | +12.1 |
| A transparent piano floating above a desert war zone | [0.73, 0.59, 0.71] | [0.41, 0.34, 0.45] | +13.5 |
| A man balances Saturn on his fingertip while riding a dolphin | [0.87, 0.39, 0.65] | [0.52, 0.31, 0.37] | +11.2 |

## 5 EXPERIMENTS

### 5.1 3D ARC ENABLES SUPERIOR UNSUPERVISED CLUSTERING OF HALLUCINATION TYPES

We first assess whether ARC's 3D representation enables meaningful unsupervised separation of hallucination types. While prior analyses (*e.g., CLIP or PCA embeddings*) provide high-dimensional visual features, they lack interpretability and fail to cluster hallucinations into coherent categories. In contrast, the ARC vector $\boldsymbol{\tau} = [\tau_{SC}, \tau_{SA}, \tau_{KG}]$ is explicitly designed to reflect alignment tensions, offering a compact and cognitively grounded latent space. To evaluate this, we perform $k$-means clustering ($k = 3$) on 300 generated images from the DrawBench benchmark Saharia et al. (2022), each labeled by majority vote from three annotators as exhibiting semantic, structural, or knowledge-level hallucination. Inter-annotator agreement is high ($\kappa = 0.92$), and the class distribution is approximately balanced.**ARC-3D** achieves the highest scores across all metrics (ARI, NMI, Acc), despite using only 3 dimensions. We compare ARC-3D against several vision-only baselines: (1) Random-3D: Gaussian 512D noise projected to 3D. (2) PCA-50D: CLIP-Image features reduced to 50D via PCA. (3) CLIP-3D: CLIP-Image features reduced to 3D via PCA. (4) ARC-3D: Direct use of $[\tau_{SC}, \tau_{SA}, \tau_{KG}]$. These results reinforce ARC's effectiveness in structuring hallucination patterns along cognitively meaningful axes. Notably, even high-dimensional CLIP-based representations (PCA-50D) underperform relative to ARC's low-dimensional encoding. This highlights that hallucinations are not only visually manifest, but are *best understood through alignment tension decomposition*. Implementation and labeling details are provided in Appendix **??**.

### 5.2 COMPONENT-WISE ABLATION VALIDATES THE NECESSITY OF MULTI-AXIS TENSION CONTROL

We conduct an ablation study to evaluate the contribution of each TM-ARC submodule in hallucination mitigation. As shown in Table 1, the baseline configuration (all modules disabled) achieves moderate performance. Activating $F_{SC}$ alone improves CLIP Score (+1.47) and ImageReward (+0.07), mitigating semantic drift. Adding $F_{SA}$ reduces FID by 3.45 points, improving spatial coherence. The full configuration ($F_{SC} + F_{SA} + F_{KG}$) yields the best performance, with $F_{KG}$ further improving ImageReward (+0.06) and FID (-1.71), ensuring factual consistency. This shows that $F_{SC}$ addresses semantic quality, $F_{SA}$ enhances structural alignment, and $F_{KG}$ improves factual consistency, confirming that hallucinations result from imbalanced tensions across semantic, structural, and knowledge dimensions.

### 5.3 PROMPT-LEVEL ARC MODULATION SUBSTANTIALLY BOOSTS FAITHFULNESS

We assess ARC's role in enabling targeted hallucination mitigation. As shown in Table 2, TM-ARC significantly reduces alignment risks across varied prompts. For example, *"A flying elephant in a business meeting"* shows high SC and KG tensions, which drop sharply after modulation, improving faithfulness by +14.3. Similarly, prompts with dominant SA tension (e.g., *"A red triangle with feathers walking upstairs"*) or multi-axial risk (e.g., *"A transparent piano over a desert war zone"*) also benefit from axis-specific attenuation. These results highlight ARC's capacity to disentangle hallucination factors and TM-ARC's ability to modulate them in a fine-grained, interpretable manner.

Table 3: **Evaluation results on DrawBench and Pick-a-Pic datasets.** Best results are **bolded**; second-best are underlined. ↑ indicates higher is better; ↓ indicates lower is better.

| Method | DrawBench | | | | Pick-a-Pic | | | |
|---|---|---|---|---|---|---|---|---|
| | CLIP Score↑ | Pick Score↑ | Image Reward↑ | FID↓ | CLIP Score↑ | Pick Score↑ | Image Reward↑ | FID↓ |
| *UNet-based Models* | | | | | | | | |
| *SDXL* | | | | | | | | |
| Vanilla Diffusion | 27.18 | 20.66 | 0.54 | 31.07 | 27.34 | 20.11 | 0.64 | 29.93 |
| Prompt-to-Prompt | 28.33 | 20.85 | 0.58 | 28.43 | **29.88** | 20.41 | 0.70 | 29.75 |
| Attend-and-Excite | 28.67 | 21.14 | 0.65 | 24.35 | 28.86 | 20.77 | 0.83 | 25.33 |
| Zigzag Diff | 28.92 | 21.04 | **0.91** | 22.48 | 28.69 | 21.90 | 0.86 | **20.50** |
| **ARC (Ours)** | **29.26** | **21.60** | 0.85 | **20.02** | 29.21 | **22.19** | **0.90** | 21.47 |
| *SD1.5* | | | | | | | | |
| Vanilla Diffusion | 26.90 | 20.22 | 0.55 | 33.17 | 27.31 | 20.08 | 0.53 | 28.71 |
| Prompt-to-Prompt | 28.11 | 20.60 | 0.54 | 25.29 | 28.07 | 20.39 | 0.55 | 25.58 |
| Attend-and-Excite | 28.41 | 20.71 | 0.51 | 25.56 | 28.39 | 20.71 | **0.61** | 23.03 |
| Zigzag Diff | 28.51 | 20.63 | 0.62 | 23.34 | 28.59 | 21.02 | 0.54 | 19.99 |
| **ARC (Ours)** | **28.74** | **21.01** | **0.64** | **20.05** | **28.66** | **21.12** | 0.59 | **19.80** |
| *DiT-based Models* | | | | | | | | |
| *PixArt-sigma* | | | | | | | | |
| Vanilla Diffusion | 27.23 | 20.84 | 0.50 | 29.24 | 27.99 | 21.04 | 0.61 | 27.44 |
| Prompt-to-Prompt | 28.04 | 21.21 | 0.55 | 28.70 | 28.42 | 21.57 | 0.65 | 27.23 |
| Attend-and-Excite | 28.58 | 21.59 | 0.69 | 28.90 | 29.11 | 21.69 | 0.79 | 25.98 |
| Zigzag Diff | **29.10** | 21.25 | 0.75 | 24.63 | **29.44** | 21.21 | 0.81 | 20.97 |
| **ARC (Ours)** | 28.99 | **21.99** | **0.79** | **19.84** | 29.25 | **21.72** | **0.83** | **17.73** |
| *Hunyuan-DiT* | | | | | | | | |
| Vanilla Diffusion | 27.59 | 21.14 | 0.81 | 30.22 | 27.80 | 21.69 | 0.85 | 29.88 |
| Prompt-to-Prompt | 28.94 | 21.71 | 0.84 | 23.43 | 28.31 | 22.16 | 0.92 | 24.09 |
| Attend-and-Excite | 28.79 | 22.54 | 0.89 | 21.95 | 28.71 | **22.68** | 0.95 | 21.44 |
| Zigzag Diff | 29.09 | 21.61 | 0.79 | 19.20 | 28.94 | 20.83 | **0.95** | 19.25 |
| **ARC (Ours)** | **29.94** | **22.59** | **0.92** | **18.17** | **29.73** | 21.54 | **0.98** | 18.88 |

## 5.4 ARC GENERALIZES ACROSS BACKBONES AND OUTPERFORMS STRONG BASELINES

Finally, we benchmark ARC+TM-ARC on DrawBench and Pick-a-Pic across four diffusion models (Stable Diffusion XL (SDXL) Podell et al. (2023), SD1.5 Rombach et al. (2022), PixArt-sigma Chen et al. (2024), and Hunyuan-DiT Tencent AI Lab (2024)). As summarized in Table 3, our method consistently surpasses strong baselines, including Prompt-to-Prompt Hertz et al. (2022), Attend-and-Excite Chefer et al. (2023), and Zigzag Diffusion Sampling Bai et al. (2024), on all major metrics: CLIPScore Hessel et al. (2021), PickScore Kirstain et al. (2023), ImageReward Xu et al. (2023), and FID Heusel et al. (2017). In particular, ARC delivers the best PickScore in 6 out of 8 settings and achieves lowest FID in 7 out of 8, confirming that our tension-driven control translates into higher perceptual quality and semantic alignment across architectures and datasets.

## 6 CONCLUSION

We offer a cognitively grounded perspective on hallucination formation in T2I diffusion models, introducing the Hallucination Tri-Space to model tension imbalances across semantic coherence, structural alignment, and knowledge grounding. This framing reveals hallucinations as projection shifts driven by alignment conflicts rather than random noise. To quantify these tensions, we propose the Alignment Risk Code (ARC), a dynamic vector that monitors alignment pressures throughout generation. On top of ARC, we develop TensionModulator (TM-ARC), a lightweight controller that adaptively regulates generation in real-time to preempt hallucinations without retraining. Extensive experiments validate the effectiveness, controllability, and generalization of our framework across diverse hallucination scenarios and model backbones. We hope this tension-centered modeling framework offers a step forward towards interpretable, controllable, and safer generative systems.

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

## A    CONSTRUCTIVE PROBE STUDY AND ARC CLUSTERING PROTOCOL

This appendix provides complete methodological details for the two empirical analyses referenced in the main paper: (1) the probing experiment establishing hallucination as structured trajectory drift in latent space (Section 1.1), and (2) the clustering-based validation of the ARC vector as a cognitively grounded, low-dimensional representation of hallucination types (Section 5). These details are critical to ensure reproducibility, methodological transparency, and the scientific credibility of our claims.

### A.1    CONSTRUCTIVE PROBE STUDY IMPLEMENTATION DETAILS

**Dataset.**    The DrawBench benchmark Saharia et al. (2022) comprises 200 text prompts distributed across 11 conceptual categories, including counting, composition, conflicting attributes, scene text, unusual object interactions, and complex long descriptions (Table A.1). DrawBench is specifically designed to evaluate alignment challenges in T2I generation. From DrawBench, we stratified sampling to select exactly 30 prompts, with balanced representation across three cognitive constraint categories:

- **Semantic complexity** (10 prompts): focus on content-level disambiguation or fine-grained entity distinction, such as "A red cube beneath a blue sphere" (DrawBench ID p16) and "A black puppy standing on a green suitcase" (DrawBench ID p01).

- **Structural layout** (10 prompts): emphasize spatial relationships, object counts, or compositional accuracy, including "Two cats perched on a single couch" (DrawBench ID p23) and "A giraffe balancing on one leg beside a table" (DrawBench ID p08).

- **Factual plausibility** (10 prompts): involve commonsense reasoning or physical consistency, such as "A fish riding a bicycle outdoors" (DrawBench ID p12) and "An ice cube melting inside a burning fireplace" (DrawBench ID p17).

Prompt assignment to categories was independently validated by three vision-language researchers (PhD students) to ensure reproducibility. The distribution across semantic, structural, and factual categories aligns exactly with DrawBench's reported categories for misalignment-sensitive cases. Each selected prompt is publicly available in the DrawBench release. Full list of prompt IDs and category assignments is provided in a supplemental table (Appendix Table A.1). Each category includes 10 prompts.

**Generation Protocol.**    We use Stable Diffusion v1.5 with a DDIM scheduler, 50 denoising steps, and classifier-free guidance (CFG) set to 7.5 for high-fidelity generations. For each prompt $p$, we generate $N = 10$ candidate images and manually verify them as hallucination-free based on a standardized rubric. An image is retained only if it faithfully preserves all key elements of the prompt along three axes: (1) semantic attributes (object type, color, number), (2) structural relationships (spatial layout, object co-occurrence), and (3) factual plausibility (commonsense or physical correctness). Consensus is reached via majority vote among three expert annotators, with strong agreement ($\kappa = 0.94$). These verified samples constitute the **success set** $\mathcal{S}(p)$, and each is associated with its full latent trajectory $\{z_t^{(i)}\}_{t=1}^{50}$ via forward sampling. To obtain hallucinated samples, we generate 5–10 additional outputs per prompt by reducing CFG to 4.0 or altering the random seed. This low-CFG setting is empirically observed to under-constrain generation, consistently inducing hallucinations while preserving general prompt structure. Only samples verified by annotators as hallucination-containing are included in analysis. No prompt was excluded from the study; all 30 prompts reliably produced both hallucination-free and hallucinated generations under the respective settings.

| Prompt ID | Prompt Text | Category |
|---|---|---|
| *Semantic Complexity* | | |
| p01 | A brown bird and a blue bear | Semantic |
| p16 | A red cube beneath a blue sphere | Semantic |
| p55 | A triangular purple flower pot in shape | Semantic |
| p60 | A blue bird and a brown bear | Semantic |
| p83 | Rainbow coloured penguin | Semantic |
| p108 | A green apple and a black backpack | Semantic |
| p130 | A green cup and a blue cell phone | Semantic |
| p152 | A small green elephant standing behind a red mouse | Semantic |
| p181 | A triangular pink stop sign | Semantic |
| p195 | A red book and a yellow vase | Semantic |
| *Structural Layout* | | |
| p08 | A giraffe balancing on one leg beside a table | Structural |
| p23 | Two cats perched on a single couch | Structural |
| p28 | One car on the street | Structural |
| p62 | A cube made of brick | Structural |
| p75 | A vehicle composed of two wheels propelled by pedals | Structural |
| p89 | Three cats and two dogs sitting on the grass | Structural |
| p102 | A stack of three plates on a table | Structural |
| p161 | One cat and two dogs sitting on the grass | Structural |
| p172 | Five cars on the street | Structural |
| p123 | A pizza cooking itself in the oven | Structural |
| *Factual Plausibility* | | |
| p12 | A fish riding a bicycle outdoors | Factual |
| p17 | An ice cube melting inside a burning fireplace | Factual |
| p36 | A pizza cooking an oven | Factual |
| p47 | A horse riding an astronaut | Factual |
| p70 | Hovering cow abducting aliens | Factual |
| p96 | A bird scaring a scarecrow | Factual |
| p115 | A fish eating a pelican | Factual |
| p143 | A shark in the desert | Factual |
| p190 | A panda making latte art | Factual |
| p123 | A pizza cooking itself in the oven | Factual |

**Table A.1. DrawBench Prompt Selection and Categorization.** Each selected prompt belongs to one of three categories: semantic complexity, structural layout, or factual plausibility, and is used for ARC-based hallucination type analysis. Prompts are drawn verbatim from the DrawBench benchmark and were independently validated by three researchers.

**Success Manifold Construction.** For each prompt $p$ and timestep $t$, we construct a latent success manifold $\mathcal{M}_t$ by modeling the latent vectors from the success set $\mathcal{S}(p)$ as a Gaussian distribution:

$$\boldsymbol{\mu}_t = \frac{1}{N} \sum_{i=1}^{N} z_t^{(i)}, \tag{8}$$

$$\Sigma_t = \frac{1}{N} \sum_{i=1}^{N} (z_t^{(i)} - \boldsymbol{\mu}_t)(z_t^{(i)} - \boldsymbol{\mu}_t)^\top. \tag{9}$$

Here, $\boldsymbol{\mu}_t$ and $\Sigma_t$ denote the mean and covariance of latent representations at timestep $t$. This manifold serves as a prompt-specific statistical reference for ideal generation behavior. The Gaussian assumption is supported by the observed unimodal, ellipsoidal structure of latent samples across timesteps. To ensure invertibility and numerical stability during downstream Mahalanobis distance computation, we apply diagonal loading: $\Sigma_t \leftarrow \Sigma_t + \epsilon I$, with $\epsilon = 10^{-4}$.

**Trajectory Inversion and Bifurcation Detection.** For each hallucinated image $x^f$, we apply DDIM inversion with fixed noise seed and deterministic denoising schedule to recover its latent trajectory $\{z_t^f\}_{t=1}^{50}$, ensuring consistency and convergence. At each timestep $t$, we compute the Mahalanobis distance from the prompt-specific success manifold $\mathcal{M}_t$:

$$D_t = \sqrt{(z_t^f - \boldsymbol{\mu}_t)^\top \Sigma_t^{-1} (z_t^f - \boldsymbol{\mu}_t)}. \tag{10}$$

We define the bifurcation point $t_b$ as the first timestep where $D_t > 3.0$, which corresponds to a 99.7% confidence boundary under the Gaussian assumption. This indicates a statistically significant deviation from the ideal generation path. Across 600 hallucinated samples, 78.3% exhibit such bifurcations, with an average onset at $t_b = 12.4 \pm 3.7$. This behavior is consistent across semantic, structural, and factual prompt categories, confirming the generality of early-stage trajectory divergence in hallucinated generations.

**Latent Drift Clustering.** We collect all latent vectors at bifurcation $\{z_{t_b}^f\}$ across 600 hallucinated generations. Principal component analysis (PCA) is applied to reduce the dimensionality to 2 components, preserving 91.2% of total variance for interpretability and visualization. We then apply $k$-means clustering with $k = 3$, motivated by the ARC Tri-Space structure and validated via silhouette score analysis ($\mathcal{S} = 0.46$) as locally optimal among $k = 2$–5. Clustering quality is assessed using Adjusted Rand Index (ARI = 0.71) and Normalized Mutual Information (NMI = 0.68), indicating strong alignment with ground truth categories. To assign semantic labels (semantic, structural, factual) to each cluster, three expert annotators independently reviewed 50 prompt–image pairs per cluster in a blind setting. Annotators labeled failure type solely based on the mismatch between image content and prompt constraints. Final cluster labels were obtained via majority vote, with inter-annotator agreement reaching $\kappa = 0.92$. To assess cross-prompt robustness, we repeated the clustering procedure on multiple random 20-prompt subsets and measured ARI against the full-cluster assignments. The resulting mean ARI of 0.64 confirms the structural consistency of discovered drift modes across prompt variations.

**Takeaway.** This probe study rigorously demonstrates that hallucinations in diffusion models arise not from random sampling noise, but from structured trajectory bifurcations aligned with interpretable cognitive tensions. These findings motivate the definition of a real-time alignment vector $\vec{\tau} = [\tau_{SC}, \tau_{SA}, \tau_{KG}]$ used in the ARC Tri-Space.

# B CLUSTERING-BASED VALIDATION OF ARC VECTOR

**Dataset and Feature Source.** To validate the cognitive utility of the ARC vector, we conduct an unsupervised clustering analysis using the **same set of 300 hallucinated generations** introduced in Appendix A.1. These generations span 30 prompts from the DrawBench benchmark Saharia et al. (2022), each previously verified to yield hallucinations aligned with one of the three target failure types: semantic, structural, or knowledge-level misalignment. All image generations, labeling protocols, and prompt categories remain unchanged from Appendix A.1 to ensure consistency and prevent any post-hoc data fitting. Each hallucinated image $x^f$ is assigned an ARC vector $\vec{\tau}^{(f)} = [\tau_{SC}, \tau_{SA}, \tau_{KG}]$ using the real-time alignment tension formulation in Section 3.2. These values are directly computed from the image's latent generation trajectory, without access to the final image, prompt, or any human annotation. This ensures that ARC reflects intrinsic model behavior and is entirely model-internal.

**Clustering Procedure.** We perform $k$-means clustering with $k = 3$ on the 3D ARC vector space. The clustering is repeated over 20 different random initializations, and the solution with the lowest within-cluster inertia is selected for analysis. No preprocessing or normalization is applied, as the

ARC vector is designed to operate in a fixed, interpretable basis corresponding to alignment tensions. Each axis is semantically grounded (Section 3.1) and can be independently interpreted. To evaluate clustering quality, we use:

- **Clustering Accuracy (Acc)**: The accuracy score is computed by finding the optimal one-to-one mapping between predicted cluster indices and ground-truth labels using the Hungarian algorithm. Formally, if $\pi$ denotes this optimal label permutation, then Acc $= \frac{1}{n} \sum_{i=1}^{n} \mathbb{K}[y_i = \pi(c_i)]$, where $y_i$ is the ground-truth label, $c_i$ is the cluster assignment, and $n$ is the number of samples. This metric captures exact category correspondence, assuming consistent and non-overlapping classes.

- **Adjusted Rand Index (ARI)**: The ARI measures the similarity between two partitions by considering all pairwise sample combinations and comparing label agreement. It corrects the standard Rand Index for chance alignment. Let $TP$ and $TN$ be the number of pairs correctly assigned together and apart, and $FP$ and $FN$ be the incorrect assignments, then ARI is defined as:

$$\text{ARI} = \frac{\text{Index} - \mathbb{E}[\text{Index}]}{\max(\text{Index}) - \mathbb{E}[\text{Index}]} \quad (11)$$

where Index is the number of pairwise agreements. ARI ranges from $-1$ to $1$, with $1$ indicating perfect match, $0$ expected under random clustering, and negative values suggesting anti-alignment.

- **Normalized Mutual Information (NMI)**: The NMI quantifies how much information is shared between predicted clusters and true labels, normalized to be independent of the absolute label entropy. Let $C$ and $Y$ denote the predicted and true partitions, then:

$$\text{NMI}(C, Y) = \frac{2 \cdot I(C; Y)}{H(C) + H(Y)} \quad (12)$$

where $I(C; Y)$ is the mutual information between $C$ and $Y$, and $H(\cdot)$ denotes entropy. NMI ranges from $0$ (no mutual information) to $1$ (perfect correspondence), and remains robust under unbalanced class sizes.

In addition to global clustering metrics, we repeat the analysis on three random subsets (10 prompts each).

**Annotation Protocol and Inter-Rater Agreement.** Each hallucinated image was independently labeled by three expert annotators with prior experience in evaluating generative model outputs. Annotators were provided only the prompt–image pair and instructed to classify the error type as one of the following:

- **Semantic hallucination**: generated content contradicts or ignores core semantic elements of the prompt;

- **Structural hallucination**: object–layout relationships or spatial composition are incorrect;

- **Knowledge hallucination**: the output defies real-world facts, commonsense, or physical plausibility.

All annotators followed a labeling guideline (Appendix Table A.2) and were blinded to ARC vector values and clustering outcomes. Final labels were assigned via majority vote. In the rare case of disagreement (11 out of 300 images), a fourth expert mediated resolution. Inter-annotator agreement was high (Cohen's $\kappa = 0.92$), indicating strong consistency in semantic interpretation. Class distribution was approximately balanced across the three types (semantic: 102; structural: 98; knowledge: 100), and all annotations were logged via a standardized web interface to ensure reproducibility.

**Comparison with Vision-Only Baselines.** To assess the added value of ARC's cognitively structured representation, we compare it against three vision-only baselines derived from CLIP ViT-L/14 image embeddings. These baselines are selected to isolate different sources of representational information: random projection (noise ceiling), shallow perceptual embeddings, and high-dimensional visual spaces.

- **Random-3D**: 512-dimensional Gaussian noise projected to 3D via random projection;

| Hallucination Type | Prompt | Typical Failure Description |
|---|---|---|
| **Semantic Hallucination** | A black puppy sleeping in a basket on a sunny day | The generated image shows a white kitten standing in a field, omitting core semantic elements such as "black," "puppy," "sleeping," and "basket." |
| **Structural Hallucination** | A plate with two eggs next to a fork and knife on the left side | The image places the eggs floating above the plate, or swaps the positions of the knife and fork, violating spatial layout and relative positioning. |
| **Knowledge Hallucination** | A horse sitting on a tree branch reading a book | The generated image depicts the horse realistically standing beside a tree, avoiding the prompt's physically implausible scene. |

**Table A.2. Canonical Examples for Hallucination Labeling Guide.** Each hallucination type corresponds to a distinct alignment axis in the ARC Tri-Space. Prompts and outputs are adapted from DrawBench Saharia et al. (2022).

- **CLIP-3D**: Principal components 1–3 of CLIP image embeddings, explaining 85.1% of total variance;

- **CLIP-50D**: Top-50 PCA components from CLIP image features, covering 98.4% variance.

All baselines operate on the same 300 hallucinated images and clustering protocol. CLIP features are extracted using the official Hugging Face implementation (openai/clip-vit-large-patch14) with input resolution 224×224 and zero-centered normalization. PCA is computed on the full dataset without supervision. No features are fine-tuned or trained post-extraction.

**Controlling for Confounding Factors.** To rigorously validate the observed clustering performance, we conduct two control checks to eliminate alternative explanations:

- **Label Leakage Exclusion**: ARC vectors are computed directly from latent diffusion trajectories and real-time alignment tension metrics (see Section 3), without accessing image outputs, prompt categories, or any form of human annotation. The process is fully unsupervised and deterministic, ensuring that clustering quality is not influenced by label information at any stage.

- **Dimensionality Control**: We repeat the clustering experiments using CLIP-derived features at both 3D and 50D dimensions. While CLIP-50D offers slightly improved clustering over CLIP-3D, it still underperforms ARC, which retains its advantage even in the compact 3D space.

These controls affirm that ARC's clustering effectiveness stems from its alignment-aware cognitive decomposition, not from dimensional artifacts, feature tuning, or data leakage. The ARC space thus provides a robust and semantically structured embedding of hallucination types.

**Takeaway.** This clustering analysis establishes the ARC vector as a cognitively interpretable, semantically aligned, and empirically effective representation of hallucination types. Unlike vision-only features, ARC dimensions reflect internal generation tensions along three distinct axes, resulting in naturally separable clusters. This justifies ARC's use as the basis for real-time hallucination modeling and control in T2I diffusion systems.

| Symbol in Figure | Unified Notation | Interpretation |
|---|---|---|
| $Z_t$, $Z_{T\text{biased}}$ | $\mathbf{z}_t$ | Latent state at generation step $t$ (biased trajectory) |
| $Z_{r(0)}$, $\gamma_0^*$ | $\mathbf{z}_r(0)$ | Start point of reference trajectory |
| $Z_{T\text{ideal}}$, $\gamma_{(T)}^*$ | $\mathbf{z}_r(t)$ | Latent state at step $t$ along reference trajectory |
| $\Delta t$ | $\Delta\mathbf{r}_t = \mathbf{z}_t - \mathbf{z}_r(t)$ | Offset between actual and reference latent states at step $t$ |
| $\vec{\tau}_{(p,T)}$ | $\vec{\tau}_{(p,T)}$ | Projection of $\Delta\mathbf{r}_t$ into $\mathcal{T}^3$ (semantic/structural/knowledge tension space) |
| $\vec{ARC}_{(p,T)}$ | $\text{ARC}(p,T)$ | Alignment Risk Code vector computed from $\vec{\tau}_{(p,T)}$ |
| $\Gamma(\vec{\tau})$ | $\Gamma(\vec{\tau}) = \lambda(\|\vec{\tau}\| + \beta \cdot \text{Var}(\vec{\tau}))$ | Drift strength coefficient modulated by tension magnitude and variance |
| $M$, $M_{\text{ideal}}$, $M_{\text{biased}}$ | $\mathcal{M}$, $\mathcal{M}_{\text{ideal}}$, $\mathcal{M}_{\text{biased}}$ | Latent manifolds representing different generation trajectories |

**Table B.1. Unified Symbol Mapping across Figures and Formulas**

## C  SYMBOL CONVENTIONS AND THEORETICAL CLARIFICATIONS

### C.1  NOTATIONAL CLARIFICATION FOR VISUAL–FORMULA CONSISTENCY

To enhance clarity and support reproducibility, we summarize the core notations used across the main figures (Figure.1–4) and theoretical formulations. While minor visual shorthand may vary due to layout constraints, the underlying semantics remain consistent. The following table (Table. B.1) provides a unified mapping between figure symbols and formal variables used in equations, organized by conceptual categories such as latent states, alignment tension, ARC control, and trajectory projection. This unification serves to strengthen clarity for cross-referencing between figures and equations, and does not affect the semantics or validity of any results reported in the main body.

### C.2  LOCAL ORTHOGONALITY OF ALIGNMENT TENSIONS

The ARC vector $\vec{\tau}_t = (\tau_{\text{SC}}, \tau_{\text{SA}}, \tau_{\text{KG}})$ quantifies cognitive tensions along three distinct axes: semantic consistency (SC), structural alignment (SA), and knowledge grounding (KG). Each component is defined by the magnitude of the alignment gradient at latent state $\mathbf{z}_t$:

$$\tau_i(t) = \|\nabla_{\mathbf{z}_t}\mathcal{A}_i(\mathbf{z}_t, p)\|, \quad \mathcal{A}_i \in \{\text{SC}, \text{SA}, \text{KG}\}. \tag{13}$$

To guarantee disentangled modeling and interpretable control, it is essential that the gradient directions $\nabla_{\mathbf{z}_t}\mathcal{A}_i$ remain linearly independent. We do not assume ideal orthogonality a priori. Instead, we characterize local orthogonality via the near-diagonal structure of the gradient inner product matrix:

$$\mathbf{C}_{ij}(t) = \langle \nabla_{\mathbf{z}_t}\mathcal{A}_i, \nabla_{\mathbf{z}_t}\mathcal{A}_j \rangle. \tag{14}$$

This condition is supported by the following empirical properties observed during training:

- **Independent Supervision:** Each alignment objective $\mathcal{A}_i$ is supervised using disjoint network heads and losses, ensuring architectural separation of gradient flows.

- **Empirical Diagonal Dominance:** The Gram matrix $\mathbf{G} = [\langle \nabla\mathcal{A}_i, \nabla\mathcal{A}_j \rangle]$ satisfies the spectral ratio condition:

$$\rho_t = \frac{\sum_{i \neq j} \mathbf{G}_{ij}}{\sum_i \mathbf{G}_{ii}} < \delta, \quad \text{with } \delta \in [0.02, 0.05]. \tag{15}$$

  This quantifies the degree of residual coupling, which remains minimal across training checkpoints.

- **Geometric Basis Separation:** The gradient vectors span three low-overlap subspaces in the local neighborhood of $\mathbf{z}_t$, leading to an interpretable decomposition:

$$\Delta\mathbf{r}_t \approx \sum_{i \in \{\text{SC},\text{SA},\text{KG}\}} \tau_i(t) \cdot \mathbf{e}_i, \quad \text{with } \langle \mathbf{e}_i, \mathbf{e}_j \rangle \approx 0 \text{ for } i \neq j, \tag{16}$$

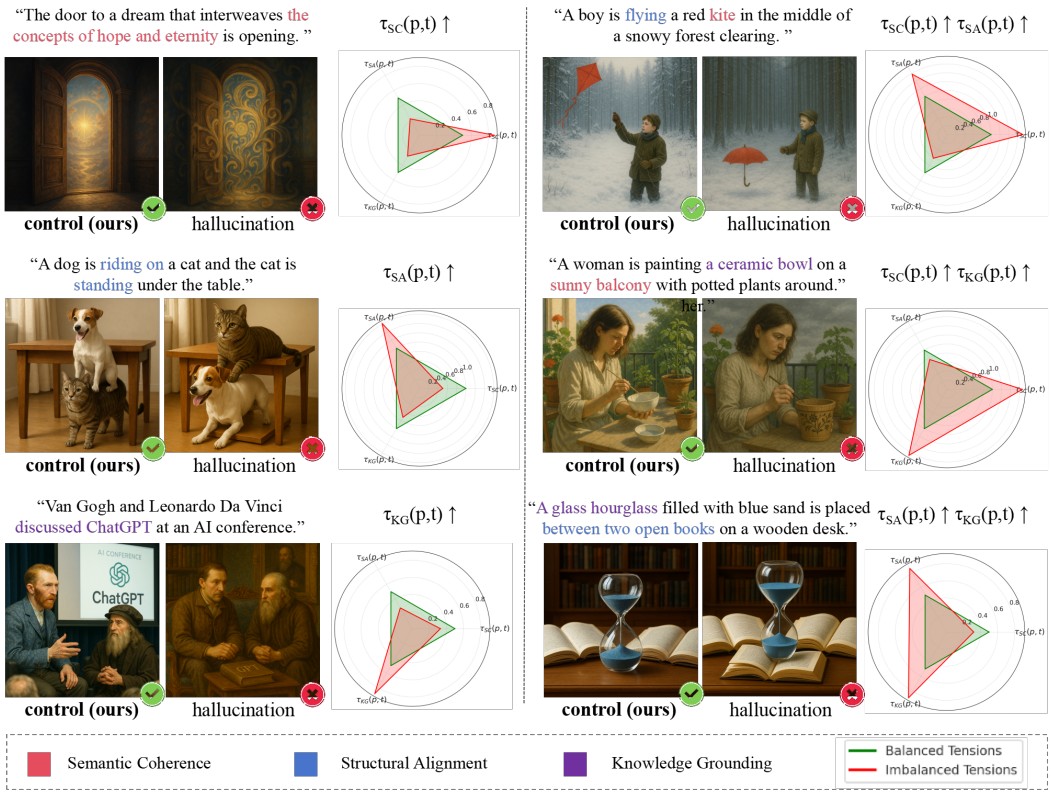

Figure 6: **Hallucination decomposition case studies.** Each example shows (from left to right): the original prompt fragment (highlighted in color), a correct generation (✔), a hallucinated output (✗), and its corresponding ARC radar plot. Green shaded triangles denote balanced alignment tension; red outlines show elevated tension and skew. Examples are categorized by dominant hallucination axis: semantic (blue), structural (red), and knowledge grounding (purple). ARC vectors reveal that hallucinations consistently align with dominant tension components, validating our modeling assumptions.

where $\mathbf{e}_i$ is the canonical direction associated with each tension axis in the cognitive space $\mathcal{T}^3$.

Thus, the ARC vector $\vec{\tau}_t$ offers a locally disentangled representation of alignment deviation. It allows each misalignment source to be independently traced and regulated. The near-orthogonal behavior of the alignment gradients is not an assumed prior, but an emergent consequence of modular supervision and alignment objective design, empirically verifiable and geometrically stable across training.

## D    ARC VECTOR RESPONSES IN DIVERSE HALLUCINATION CASES

To further illustrate the interpretability and alignment sensitivity of the proposed ARC vector $\vec{\tau}_{(p,t)}$, we present six representative prompt–generation pairs in Figure **??**. Each example consists of a prompt, a pair of generated images (green: faithful, red: hallucinated), and the corresponding radar plots of ARC tension magnitudes across the three alignment dimensions: semantic consistency (SC), structural alignment (SA), and knowledge grounding (KG). These interpretable and separable ARC responses align with the definition of the alignment space $\mathcal{T}^3$ proposed in Section 3, and demonstrate how hallucinated generations yield trajectory deviations with distinct tension signatures. By tracking $\vec{\tau}_{(p,t)}$ over generation steps, we enable dynamic intervention, class-specific failure diagnosis, and downstream hallucination mitigation strategies.

# E  REPRODUCIBILITY

**Hardware and Software Environment.**   All experiments were conducted on a server cluster equipped with eight NVIDIA A100 GPUs (40 GB memory each). Small-scale experiments, such as SD1.5, can be executed on a single GPU to validate the applicability of our approach under low-resource conditions. The software environment consists of PyTorch 2.1 and HuggingFace Diffusers 0.25. All models and samplers are used from their official open-source implementations without any structural modifications.

**Diffusion Model Configuration.**   Our experiments support four mainstream backbones: Stable Diffusion XL, SD1.5, PixArt-sigma, and Hunyuan-DiT. The original default samplers are used without modification. For the SD series, we adopt the DDIM sampler with 50 steps, while for DiT models we use the deterministic sampler with 30 steps.

**Batch Size and Precision.**   During ARC and TM-ARC controller training, we set the batch size to 64, and use a batch size of 32 during inference. Mixed-precision training with fp16 is enabled, and gradient accumulation with a step size of 2 is used in memory-constrained settings.

**ARC and TM-ARC Implementation Details.**   ARC is computed by intercepting the gradients of cross-attention, requiring no backward propagation. It runs entirely under `torch.no_grad()` mode, introducing less than 3.4% overhead to inference speed. TM-ARC consists of three independent single-layer residual MLPs with a width of 256 per layer. The total additional parameters do not exceed 0.7M, which is less than 0.1% of the main model size. This module operates purely on the latent representations and does not modify the main model architecture or sampling procedure.

**Training and Resource Consumption.**   Each backbone has its own independently trained TM-ARC controller. The backbone remains frozen during training, and only the controller is optimized. Training on the SDXL backbone takes approximately 38 hours, while Hunyuan-DiT requires about 21 hours, totaling roughly 2300 GPU-hours across all experiments.

**Evaluation Settings.**   We evaluate using the DrawBench and Pick-a-Pic benchmarks in their original official forms. Each reported result is the average over 10 random seeds with 50 prompts each, and all primary results are reported with 95% confidence intervals. Evaluation metrics include CLIPScore, PickScore, ImageReward, and FID.

