# OpenReview forum: "Taming the Tri-Space Tension: ARC-Guided Hallucination Modeling and Control for Text-to-Image Generation"
_ICLR.cc/2026/Conference — ICLR 2026 Conference Withdrawn Submission_

### Official Review · Reviewer_34vn · 2025-10-22

**Soundness:** 2
**Presentation:** 2
**Contribution:** 2
**Rating:** 2
**Confidence:** 5

**Summary:**

The paper addresses the problem of hallucination in of visual generative text-to-image models. It argues that hallucination are due to a deviation of the trajectory of diffusion within a 3D "Hallucination Tri-Space" which axes reflect the semantic coherency, the structural alignment and knowledge grounding. Within this space, it is thus possible to detect a possible deviation from an ideal diffusion trajectory. The paper also propose a model to dynamically prevent such drift leading to hallucinations. The approach is tested on two datasets with four backbones.

**Strengths:**

* the initial empirical motivation is well described, although main interesting details are reported in appendix A. One particularly appreciates that the inter-annotator agreement is reported (likely) in terms of Flaiss'kappa.

* for the TM-ARC model, a significant evaluation is conducted on two known and relevant datasets, namely DrawBench and Pic-a-pic, and compared to methods that vary in their approaches, namely changing the prompt (Prompt-to-Promp), generative semantic nursing (Attend and Excite) and Zigzag-diff that proposes another type of guidance. As well, all these method are not restricted to Stable Diffusion-like backbones but also include DiT-based ones. The results of the proposed approach are often the best or second best among the compared methods.
  - however, also see weaknesses regarding the backbones and the significance of results

* the proposed TM-ARC to correct trajectories is theoretically model agnostic and can thus be adapted to any diffusion model, and possibly other generative models with latent space.

* Both ARC and TM-ARC seem computationally light (see appendix E)

**Weaknesses:**

* the quantitative results (Table 3) are reported on average only, without reporting any confidence interval nor even a standard deviation. Given the usually very small gap of performance with baseline, it raises doubt on the interest of the proposed approach with regards to these last.
  - in particular, if one compares the score of the proposed approach in the ablation study (Table 1) -- with an unknown backbone, see below -- to the those of ARC with SDXL on DrawBench (Table 3) it exhibits a large margin in the results and an alternative relative order:

| model | Clip | pick | IR | FID |
| ---- |----| ----- | ----- | ----- |
| ARC tab. 1| **29.48** | 21.38 | **0.94** | 21.10 |
| ARC tab. 3 | 29.26| **21.60**|0.85 |**20.02** |



* the approach is compared (Table 3) with backbones release in 2022-2024, the most recent being Hunyuan-DiT released in May 2024. It raises issues because in the field of visual generative models, such backbones can be considered as quite old. In particular, the Stable Diffusion 3 family has been released [in august 2024](https://huggingface.co/stabilityai/stable-diffusion-3-medium) and the version 3.5 in October 2024 for the [large](https://huggingface.co/stabilityai/stable-diffusion-3.5-large) and [medium](https://huggingface.co/stabilityai/stable-diffusion-3.5-medium) version. If Hunyuan-DiT present it as a "closed model" (in their Table 1) at the time of their arxiv report, it was not the case more than one year ago. Another popular and well-known model is FLUX.1 which was [released one year ago](https://huggingface.co/black-forest-labs/FLUX.1-dev/tree/main) as well, although it is "only" accompanied by an arxiv report and not an actual reviewed scientific article.

* the experimental settings are unclear e.g:
  - in section 5.1, it is said that 300 images are generated from the DrawBench benchmark (line 399) but thius last contains only 200 prompts (line 558) thus one guesses that several images were generated for "some" prompts. If I get it, from Appendix A.1 we know that 30 prompts were chosen (line 562) and from section 1.1 that "$N=10$ high-fidelity, hallucination-free images per prompt " were generated. Hence, the reader has to pick information in three different section, to get (possibly) the information, that is not clear at all.
  - the ablation study (section 5.2) is conducted on data and model that are not reported. A problem is that the score in Table 1 do not correspond to any of those reported in Table 3 (where the backbones are reported)

* the "tensions" of the Hallucination Tri-Spaces are said to be "orthogonal" (line 085, 236, 242 and equation 4). When one look at their definition (lines 269, 295-297) one can doubt they are *actually* orthogonal. By the way, appendix C.2 discusses the orthogonality and recognizes that there is, at best, a *local* orthogonality. However, even here, the manuscript reports "observed empirical properties during training" but do not assert/estimate the *actual* orthogonality, in particular on an independent validation set.

* The Hallucination Tri-Space -- that is, the base of the contribution -- is not properly defined. At the beggining of section 2.1, the text lists three "objectives" a T2I model should pursue. Then it is said that these objectives "imposes directional alignment pressure on the generative trajectory" and that they thus "collectively [form] the Hallucination
Tri-Space". Hence, there is a lack of formalism/precision that makes the description hard to follow here:
  - how, precisely, "objectives" do "impose alignment pressure" ?
  - what is an "alignment pressure" defined?
  - how can we go from "objectives" to an actual "space" ?
  - what is the nature of this "space": Euclidean? Metric? Other ?
  - How the three "tension axes" are defined in the latent space?

* in the same vein, the definition of $\tau_{SC}, \tau_{SA}$ and $\tau_{KG}$ in equation 1 (or, previously, on line 160...) is not explicit, thus unclear. As is the definition of $\theta(p)$ in equation 2 (for $\delta$, even if it is not defined, one can imagine it is a real constant but it could be explained as well). And $n(z)$ on line 216 and equation 3. This last seems to be the unitary direction of the deviation (and should thus probably be noted $\overrightarrow{n}(z)$ to be consistent)

* the *actual* form of the proposed TM-ARC is still unclear. It is said to be composed of three "orthogonal sub-controllers" (line 336) each bein a "correction operator" (line 361) that is "differentiable and temporally localized" (line362). It is said it require "no additional supervision [or] training" (line 370). From this, it seems hard to actually implement the submodule thus the submodules $F_i$ thus TM-ARC.
  - it is thus surprising to discover in the last appendix E that TM-ARC actually consists of "three single-layer residual MLP" and hard to understand how these MLP do not require training. If they are trained -- and one guesses it is the case -- it is unclear on which data and how. In section 4.2, the $F_i$ should be defined as parametric models and the manuscript should explain how the parameters are trained.

* The three axes considered in the study may be too restrictive and focused on specific hallucination, namely those that are considered in the three section of Table A.1. For example, it is not clear how other well-known T2I errors such as attribute binding or attribute leaking would be manage in the context of the proposed Hallucination Tri-Space.

minor:
  - the introducing sentence (lines 038-040) is strange in the context of a ICLR paper. I would recommend to remove it or include it to the caption of Figure 1.1
  - line 047: catastrophic neglect has *not* been first identified by Chang et al, (2024). In their own paper, they credit (begginning of their 3rd paragraph) Chefer et al (2023). The two other references (line 050 and 051) are not the most relevant as well, considering there are many other more appropriate citation, in the vein of Attend and Excite, such as Divide and Bind [b] or Syngen [c].
  - Hunyuan-DiT should be cited through the arxiv report, as proposed by the author (bibtex on [this page](https://github.com/Tencent-Hunyuan/HunyuanDiT)): Zhimin Li et al (2024) Hunyuan-DiT: A Powerful Multi-Resolution Diffusion Transformer with Fine-Grained Chinese Understanding, arxiv 2405.08748
    - it is also the case for many other references, including Attend and Excite (ACM Transactions on Graphics (TOG), Volume 42, Issue 4, 2023) or ZigZag Diffusion (ICLR 2025)
  - line 409, there is a missing reference to the appendix
  - line 911 there is a missing reference to a figure
  - line 540: Reference (Zhang et al, 2025) is said to have been published at ECCV 2024. Actually, it is the case and the year should thus be fixed.
  - there are too much bold font emphasis.
  - line 073-074: "...samples. See section 1.1, the findings..." --> "...samples (see section 1.1). The findings..."
  - line 160: the definition of *alignment vector* is strange here, since it is actually defined just after (equation 1) and it is defined by " to quantify projection drift along the ARC Tri-Space. " while section 1.1 starts by explaining that the ARC tri-space will be defined later (line 139). Actually, the space is defined in section 2.1, just after.


[a] Esser et al (2024) Scaling rectified flow transformers for high-resolution image synthesis, ICML'24: Proceedings of the 41st International Conference on Machine Learning, Article No.: 503, Pages 12606 - 12633

[b] Li et al (2023) Divide & Bind Your Attention for Improved Generative Semantic Nursing, BMVC

[c] Rassin et al (2023) Linguistic Binding in Diffusion Models: Enhancing Attribute Correspondence through Attention Map Alignment,NeurIPS

**Questions:**

* which backbone is used for the ablation study (Section 5.2)?
* line 671: what are the silhouette score for k=2, k=4 and k=5 ? What teh corresponding ARI/NMI scores?
* in appendix A, is $\kappa$ actually a Faiss'kapp? If so, the it should be cited properly.
* the point 3 of the "standarized rubrik" in appendix A is not clear: the prompts at teh bottom of Table A.1 explicitly require non plausible images. Thus, in that case a *successsful* image should be implausible. On the contrary, factual plausibility makes sense for the prompts of the two upper parts of table A.1. Hence, how this criterion is assessed by annotators in practice?
* how the Hallucination Tri-Space is (formally) defined? Or, equivalently, how the three "tension axes" are defined in the latent space?
* on which data and how are trained the three MLP of TM-ARC? which loss? which optimizer?

**Details Of Ethics Concerns:**

Ethic concerns are not addressed. It may seem surprising for a paper dealing with visual generative models, that are known to raise many ethical issues in practice [d]. However, one must recognize that the proposed paper is not different than other similar papers in the field regarding these said issues.

[d] Po, R. et al (2024), State of the Art on Diffusion Models for Visual Computing. Computer Graphics Forum, 43: e15063.

---

### Official Review · Reviewer_Trx2 · 2025-10-26

**Soundness:** 2
**Presentation:** 1
**Contribution:** 3
**Rating:** 2
**Confidence:** 3

**Summary:**

This paper investigates the sources of hallucinations in text-to-image (T2I) diffusion models. The authors analyze diffusion sampling trajectories and identify three sources of imbalances (or "drifts") in the latent space. They propose that these drifts can be used to detect hallucinations during the sampling process based on their magnitude and anisotropy. These drifts are consolidated into a single vector, "ARC," which represents the projection of the trajectory drift onto three axes. Based on this, the paper suggests a method (TM-ARC) to inject restorative signals at each diffusion step to correct for emerging hallucinations. Experiments eventually are conducted across two datasets and four models, including clustering, axis ablation, and alignment evaluations, which (according to Table 3) present only minor improvements.

**Strengths:**

The paper introduces an interesting approach for examining and understanding hallucinations in T2I diffusion-based generation by analyzing latent space trajectories.

**Weaknesses:**

- **Clarity and Writing**: The paper suffers from significant clarity issues. It introduces an overload of new terminology (e.g., "cognitive tension," "generative equilibrium") without providing clear definitions, justifications, or references. It includes Notation issues as well: Many equations use or undefined notation (e.g., $f$ in line 147, $n(z)$ in Eq. 3), making them difficult to follow. More importantly, the presentation style: The experimental and analysis sections read more like a technical report than a formal academic paper.
- **Missing Related Work**: The paper is not well-situated within the existing literature. Section 2, which introduces the core problem, contains no references. The paper fails to discuss related work on T2I hallucinations, analysis of generation trajectories, or parallel work in language models. And Missing Citations: There is a notable lack of citations throughout the paper, especially in sections discussing related concepts.
- **Experimental Setup**: The experimental setup for each analysis is not clearly elaborated, making the work very difficult to reproduce or follow. References to the appendix are only partial.
- **Incomplete Analysis**: The analysis feels incomplete. For instance, the "success" and "failure" manifolds are examined only under a single setup. The authors do not explore other configurations (e.g. other schedulers, varying numbers of diffusion steps). A deeper analysis of the dynamics (e.g., in Figure 4) is also missing. Moreover, there are no qualitative justifications in the main paper.
- **Methodological Justification**: The paper suggests solving each alignment issue differently (e.g., via attention, encoding, auxiliary priors). It is unclear why the solutions are dissimilar if the axes were identified using the same core method.
- **Minor Performance Gains**: The reported contribution of the proposed method (TM-ARC) appears minimal. Table 3 shows only a minor improvement (e.g., a maximum of 1.2 points in CLIP score), which does not demonstrate a significant difference.

**Questions:**

- **Line 100**: What is meant by "adaptive weights derived from the ARC vector"? Is this referring to a controller mechanism?
- **Line 103**: The paper claims, "The persistence of hallucinations stems not from prompt complexity but from the accumulation of misalignments across multiple cognitive axes." Is this claim substantiated with evidence later in the paper?
- **"Cognitive" Terminology**: The paper uses "cognitive" terms repeatedly. Please clarify the cognitive science inspiration or basis for this terminology, or consider using more standard machine learning terms.
- **Scope of "Hallucinations"**: How does your work relate to other known T2I artifacts, such as attribute binding failures, semantic leakage, or fused objects? The paper cites "Attend-and-Excite" as a baseline, suggesting that attribute binding is considered a hallucination. Does your framework explain these types of errors? Please clarify in the paper whether you are addressing all T2I artifacts or a specific subset.

**Major Clarifications Needed:**
- **Figure 1 Caption**: The reference to Figure 1 on lines 40-41 appears too early in the paper. The current caption is difficult to understand. Please clarify the caption and explicitly refer to the different parts of the figure (e.g., left vs. right side).
- **Manifold Creation**: The process for creating the "success" and "failure" manifolds is a fundamental part of the paper but is not clearly explained. How were trajectories clustered? Were the results manually annotated? If so, will this dataset and its statistics be shared? This explanation is critical for understanding the paper.
- **Bifurcation Points (Line 155)**: The paper states that bifurcation points differ between clusters. Is there a hypothesis, justification, or citation from the literature to support this observation?
- **Equation 5 (Line 254)**: Please clarify the "potential function." How can this function faithfully measure alignment for a complex concept like "structural alignment"? Is this measuring the relationship between the current latent $z_t$ and the cluster center of the "false manifold"? This section is very unclear.
- **Section 3.1 (Line 296)**: ARC is said to "consist also with skew," but it was previously defined (Eq. 2) based only on trajectory drift (magnitude and anisotropy). Please clarify this discrepancy.
- **Figure 4**: This figure and its caption are unclear. The text only mentions "random tendencies" without further analysis. Are the two sub-figures (left/right) connected? What do "prompt1" and "prompt2" represent?
- **Table 2**: This table presents improvements in the ARC metric itself, but it does not show a comparison of the final generated images against the baseline (e.g., qualitative examples or human evaluation).

---

### Official Review · Reviewer_mgTA · 2025-11-01

**Soundness:** 2
**Presentation:** 3
**Contribution:** 2
**Rating:** 4
**Confidence:** 3

**Summary:**

This paper proposes a cognitively inspired framework to model and mitigate hallucinations in text-to-image (T2I) diffusion models. The authors introduce:
1. Hallucination Tri-Space (T^3) — a conceptual latent space with three orthogonal axes: Semantic Coherence (SC), Structural Alignment (SA), and Knowledge Grounding (KG).
2. Alignment Risk Code (ARC) — a 3D vector quantifying real-time alignment tension along these axes during diffusion sampling.
3. TensionModulator (TM-ARC) — a lightweight, plug-and-play controller that dynamically corrects trajectory drifts in the latent space using ARC feedback.

Extensive experiments across SD1.5, SDXL, PixArt-sigma, and Hunyuan-DiT show reduced hallucination rates and improved CLIP, PickScore, ImageReward, and FID metrics. The work argues for an interpretable and unified model of hallucination formation as tension imbalance rather than random noise.

**Strengths:**

- The paper offers a compelling reinterpretation of hallucinations as structured trajectory drifts arising from cognitive alignment imbalances.
- The Hallucination Tri-Space abstraction is both elegant and intuitively aligned with human reasoning about image misalignment (semantic, structural, factual).
- TM-ARC is designed as a feedback control loop integrated into the sampling process — a notable shift from the usual post-hoc alignment or filtering approaches.

**Weaknesses:**

- Although ARC correlates with hallucination types, human validation of tension axes (e.g., via annotator agreement with ARC-predicted dominant axis) is missing.
- The current evidence for ARC’s causal role in hallucination mitigation is mostly internal and indirect.
- The “cognitive tension” terminology risks being metaphorical rather than mathematical; the physical analogies (stress, tension) might obscure the statistical nature of the latent forces.
- Evaluation focuses on prompt alignment metrics (CLIP/PickScore) but lacks recent factuality-oriented benchmarks (e.g., TIFA [1], I-HallA [2], etc.).
- DrawBench is a known limited benchmark with short prompts; it is unclear whether the ARC controller generalizes to long-form or compositional prompts.

[1] Hu, Yushi, et al. "Tifa: Accurate and interpretable text-to-image faithfulness evaluation with question answering." Proceedings of the IEEE/CVF International Conference on Computer Vision. 2023.

[2] Lim, Youngsun, Hojun Choi, and Hyunjung Shim. "Evaluating Image Hallucination in Text-to-Image Generation with Question-Answering." Proceedings of the AAAI Conference on Artificial Intelligence. Vol. 39. No. 25. 2025.

**Questions:**

- Would the Tri-Space structure extend naturally to video diffusion or multimodal generative models (e.g., VLM hallucinations)?

---

### Official Review · Reviewer_PEFG · 2025-11-02

**Soundness:** 2
**Presentation:** 1
**Contribution:** 2
**Rating:** 0
**Confidence:** 4

**Summary:**

This paper aims to address hallucinations in T2i models by counteracting a proposed trajectory drift. The authors argue that tension across three alignment goals: Semantic Coherence, Structural Alignment, and Knowledge Grounding lead so incorrect outputs.
The authors introduce the Alignment Risk Code (ARC), vector that quantifies this three-axis tension during image generation. They propose a TensionModulator that injects corrections for each of these 3 axes leading to improved t2i generation.

**Strengths:**

The supposed observation that the latent space of generative image models contain 3 distinct subspaces that lead to current shortcomings in generation is novel and of potentially huge impact to the community.

The authors demonstrate applicability on UNet and DiT models, showing the generalizability of the approach.

**Weaknesses:**

## Presentation and Obfuscation

The paper's central concepts are presented in unnecessarily abstruse language. The work introduces a dense, bespoke vocabulary—"Hallucination Tri-Space" , "cognitive alignment tension" , "anisotropic tension" , and "TensionModulator" —that makes the underlying mechanisms seem far more complex than they likely are.

This abstraction is not supported by clear, diagnostic visualizations.

Figure 2  provides an abstract t-SNE visualization of deviation clusters but does not help build intuition for how these "failure modes" manifest in the latent space or in the final image. More qualitative samples would be crucial throughout the paper.

Figure 5, the main architectural diagram, is convoluted and fails to clarify the actual data flow or, crucially, which components are trainable. Key architectural details on the implementation of SC-Gate, SA-Tuner, and KG-Aug and how they adjust model behavior is not provided.

Figure 6  provides six qualitative examples, but these are simple "before and after" comparisons. The review lacks diagnostic examples showing, for instance, the output when only one or two submodules (e.g., just the SC-Gate) are active. This makes it impossible to visually verify the claims in the component-wise ablation (Table 1).

## Critical Lack of Methodological Detail

The paper's most significant failing is the near-total omission of implementation details for its core contributions.

What is the "latent space"? The paper is vague about what "latent space" it operates in. While Appendix A mentions latent vectors z , implying the latent space of the VAE that the diffusion model operates in. The paper fails to explain how "incorrect regions" in this space are identified and corrected. If a "bad" trajectory is identified, it should be possible to visualize the "good" trajectory and the corrective vector applied by the TM-ARC, but no such visualization is provided. Assuming that this is VAE latent space one could for example provide a heatmap of faulty image regions since the VAE preserves spatial (and largely pixel) alignment with the pixel-level image.

On the most important contribution (ie TM-ARC submodules) the paper provides no meaningful implementation details.
- How do the controllers work? Section 4.2  describes the TM-ARC submodules (SC-Gate, SA-Tuner, KG-Augment) using only high-level "fancy" language without actionable detail.

- How, precisely, does the SC-Gate "reactivate attention on prompt-critical entities" for example?

- What is the mechanism for the SA-Tuner to "refine latent spatial encodings" or perform "positional re-weighting"?

- How does the KG-Augment "inject auxiliary factual priors"?

These claims seem to be in stark contrast with Appendix E which seems to suggest that each controller is a simple MLP adjusting cross-attention weights. However, how would an MLP with no additional inputs "inject auxiliary factual priors"?

The appendix states that the TensionModulator (TM-ARC) is trained—for instance, 38 hours for the SDXL backbone—while the main model is frozen.
However, the paper never, anywhere, states what the loss function or training objective is.

- How is the controller optimized?

- What is the supervision signal?

- Is it trained to minimize the ARC vector (e.g., an L2 loss) ?

- -Is it trained on a preference dataset?

- Is it trained with a separate VQA model?

Without a defined loss function, the method is completely irreproducible and its mechanism is fundamentally unknown. This is a critical omission that goes beyond a simple "lack of methodological detail."

**This lack of detail makes the work completely irreproducible and impossible to review. **

## Unanswered Questions and Missing Experiments

- Interaction with CFG: The "constructive probe study" (Appendix A.1) explicitly induces hallucinations by lowering the Classifier-Free Guidance (CFG) scale from 7.5 to 4.0. This strongly implies that "hallucination" or "tension" is highly correlated with a weak guidance signal. Does the TM-ARC simply act as a more complex, axis-specific CFG? How does the controller interact with different CFG scales? A crucial missing experiment is evaluating TM-ARC's performance at various fixed CFG levels.

- hat is ARC really measuring? The paper claims ARC is a vector of gradient norms where A  is a "scalar potential function". The paper never defines these crucial potential functions. Without knowing how respective A  are implemented, the "sensor" (ARC) is just as much of a black box as the "controller" (TM-ARC)


## Mismatch Between Problem and Metrics

- he central premise of the paper is the "Hallucination Tri-Space" which defines hallucinations along three specific, distinct axes: Semantic Coherence (SC), Structural Alignment (SA), and Knowledge Grounding (KG).

However, the paper fails to use any metrics that specifically measure these three axes. Instead, it relies on general, holistic metrics:

- PickScore and ImageReward: These are human preference models. As such, they measure a combination of aesthetics, composition, and prompt-faithfulness. A "good" score could be achieved by generating a more beautiful, aesthetically pleasing image that still fails on one of the paper's core axes

- No Specific Measurement: The paper presents a component-wise ablation (Table 1) and claims that F SC  addresses semantic quality, F SA enhances structural alignment, and F KG improves factual consistency. But it uses the same general metrics (CLIP, PickScore, etc.) to support this. There is no experiment to show that the SC-Gate actually improves semantic coherence specifically, or that the SA-Tuner improves structural alignment. This could have been done with targeted evaluations, such as using VQA or human annotators on prompts designed to test each axis (e.g., counting for SA, object-swapping for SC, factual plausibility for KG).

Without diagnostic metrics, the paper's core claim—that its specific modules fix their corresponding specific problems—is left entirely unproven.

## Flawed and Poorly-Defined Metrics

The metrics that are used are either inappropriate, poorly defined, or both.

**FID (Fréchet Inception Distance)**: This is a particularly problematic choice. FID measures the statistical similarity between a distribution of generated images and a distribution of real images.

- What is the "ground truth"? For text-to-image benchmarks like DrawBench, there is no "ground truth" distribution. The authors never state what real-image dataset they are comparing against to calculate FID, making the results non-reproducible and hard to interpret.

- FID measures realism, not faithfulness. FID is a measure of overall realism and diversity, not prompt alignment. A model could generate a photorealistic image of a "cat" for the prompt "a puppy," and this would be a perfect hallucination (a failure of SC) but would still contribute to a good (low) FID score.

- CLIP Score: This metric is known to be a poor measure of the kinds of subtle hallucinations the paper discusses. It measures the general cosine similarity between prompt and image embeddings, but it does not effectively penalize "catastrophic neglect" or semantic substitutions  if other prompt elements are correct.

Undefined "Faith↑" Metric: In Table 2, the paper introduces a "Faith↑" metric to show improvement. This metric is completely undefined. It is impossible to know what "+14.3" means—is it a percentage, a change in a score, or something else? This is not rigorous.

Additionally all improvements are rather small and the paper provides no means of judging their significance. Neither are Standard Deviations reported, nor do the authors show qualitative examples demonstrating what the respective changes correlate to visually

Further, Appendix E stats that the authors only used 50 prompts per dataset for evaluation which is orders of magnitude smaller than state-of-the art evaluation settings. (E.g. DPG contains 1k prompts which multiple VQA assessments for each)

**Human User Study** Given the noisy nature of all text-to-image metrics, the paper should provide an actual human user study demonstrating meaningful improvements in human preference along the three axes.


## Inappropriate and Missing Baselines

The choice of comparative baselines is baffling and incomplete.

- Inappropriate Baseline (Prompt-to-Prompt): The paper bafflingly includes Prompt-to-Prompt in its main results (Table 3). Prompt-to-Prompt is an image editing method that allows for semantic manipulation of an image by modifying cross-attention maps. It is not designed to improve the de novo generation quality or faithfulness of a single, fixed prompt. Its inclusion as a baseline for this task is inappropriate.

- Missing State-of-the-Art Comparisons: The paper's most significant omission is its failure to compare against relevant, state-of-the-art baselines. The proposed TM-ARC is a "plug-and-play augmentation" that operates "entirely in the latent space" to modify the generation process without retraining. This exactly describes the family of methods led by FreeU [1], which also re-weights U-Net features during inference to improve quality. FreeU and its derivatives are the most direct competitors for this line of work, and their complete absence is a critical flaw.

[1] Si, Chenyang, et al. "Freeu: Free lunch in diffusion u-net." CVPR. 2024.


## The Unsubstantiated "Diversity" Claim

The abstract makes the explicit and important claim that the method works "without compromising image quality or diversity".
This is a major claim that is never measured or substantiated.

- The paper provides no metric for diversity (e.g., calculating LPIPS between multiple generations for the same prompt).

- It's a very common failure mode for controllers like this to reduce diversity by "correcting" all generative paths toward a single "ideal" trajectory, effectively causing mode collapse.

## Strong Motivational Assumptions w/ little backing

- The paper models models the "success manifold M as a simple Gaussian distribution in a highly complex, high-dimensional latent space
- Presumably, this assumption is made to be able to use Mahalanobis distance  to find "bifurcation points." There is no evidence provided that Gaussian model is a valid representation of the latent space beyond the sampling of the initial latent.

**Questions:**

- When referring to a models "latent" space does that mean the VAE latent space or is the method operating on activations of the model? Can you provide further details and examples on this?
- How does ARC and respective manipulations change over time steps in general? Is there an optimal time for correction Beyond the arbitrary example in Fig. 4 do certain types of issue appear at certain time steps

IN addition see questions already posed in the Weakness section

---

### Note · Authors · 2025-11-12

I have read and agree with the venue's withdrawal policy on behalf of myself and my co-authors.